# Machine Learning-Based 30-Day Hospital Readmission Predictions for COPD Patients Using Physical Activity Data of Daily Living with Accelerometer-Based Device

**DOI:** 10.3390/bios12080605

**Published:** 2022-08-05

**Authors:** Vijay Kumar Verma, Wen-Yen Lin

**Affiliations:** 1Department of Electrical Engineering, Center for Biomedical Engineering, Chang Gung University, Tao-Yuan 33302, Taiwan; 2Division of Cardiology, Department of Internal Medicine, Linkou Chang Gung Memorial Hospital, Tao-Yuan 33305, Taiwan

**Keywords:** COPD, readmission prediction, physical activity, activity index, machine learning, hospital readmission, COVID-19

## Abstract

Chronic obstructive pulmonary disease (COPD) is a significantly concerning disease, and is ranked highest in terms of 30-day hospital readmission. Generally, physical activity (PA) of daily living reflects the health status and is proposed as a strong indicator of 30-day hospital readmission for patients with COPD. This study attempted to predict 30-day hospital readmission by analyzing continuous PA data using machine learning (ML) methods. Data were collected from 16 patients with COPD over 3877 days, and clinical information extracted from the patients’ hospital records. Activity-based parameters were conceptualized and evaluated, and ML models were trained and validated to retrospectively analyze the PA data, identify the nonlinear classification characteristics of different risk factors, and predict hospital readmissions. Overall, this study predicted 30-day hospital readmission and prediction performance is summarized as two distinct approaches: prediction-based performance and event-based performance. In a prediction-based performance analysis, readmissions predicted with 70.35% accuracy; and in an event-based performance analysis, the total 30-day readmissions were predicted with a precision of 72.73%. PA data reflect the health status; thus, PA data can be used to predict hospital readmissions. Predicting readmissions will improve patient care, reduce the burden of medical costs burden, and can assist in staging suitable interventions, such as promoting PA, alternate treatment plans, or changes in lifestyle to prevent readmissions.

## 1. Introduction

COPD is an important disease of concern, and the Global Initiative for Chronic Obstructive Lung Disease (GOLD)-2022 committee has reported [1] that 384 million people are affected by COPD, and more than 3 million people die of COPD each year, making it the third leading cause of death. Patients with COPD have the highest number of 30-day hospital readmissions compared with other diseases, such as pneumonia, gastrointestinal disorders, heart failure, and other respiratory/ventilatory diseases [2]. Hakim et al. evaluated the accuracy of the LACE index (length of stay, acuity of admission, comorbidities, and emergency department visits within the last 6 months) to predict 30-day hospital readmissions and concluded that 56% of patients with COPD require readmission within 30 days after being discharged from hospital [3]. To realize value-based patient care, Tina Shah et al. reported [4] that one in five patients with COPD require hospital readmission within 30 days of discharge. In the long term, COPD-related hospital readmissions result in economic burdens to the patient and add a strain on the available medical resources. To reduce economic burdens and develop effective strategies to prevent readmissions, Amalakuhan et al. [5] and Goto et al. [6] used machine learning (ML) models to predict and prevent hospital readmissions using data from patient demographics, patient characteristics, and different self-reported and assessed scores, such as the Barthel index and the Charlson comorbidity index, etc. Predicting hospital readmissions may also create an opportunity to stage appropriate interventions, such as pulmonary rehabilitation programs, to help reduce the total number of readmissions and rates of morbidity and mortality. To promote such interventions, Thorpe et al. conducted semi-structured interviews with 28 patients with COPD to explore their perspectives in terms of the barrier and enabling factors associated with patient participation in pulmonary rehabilitation programs [7].

Physical activity (PA) data recording and monitoring has become an important area of study for patients with chronic diseases such as COPD, and it is strongly associated with the patient’s health status and disease progression [7,8,9,10,11]. Compared with age-matched healthy individuals or patients with other chronic diseases, -“physical inactivity”- is very common among patients with COPD [12]. The Official European Respiratory Society [13] reported that lower PA is associated with increased morbidity and mortality. Prieto-Centurion et al. [14] found that PA in daily living is very important for assessing the health condition of patients with COPD. Moreover, other studies [15,16,17] have proven that the health status of patients with COPD and acute exacerbations are highly correlated with their PA of daily living. With respect to disease progression, the PA of patients with COPD is lowered, and PA data have been used to investigate the relationship between the clinical characteristics of patients and the diseases progression [18,19,20]. In addition, different statistical PA data analysis techniques have been standardized by recording PA data from 57 patients with COPD patients using wearable devices [21].

The health status of patients with COPD can be monitored by the hospital staff through regular clinical observations and using PA data after discharge from the hospital. Using an accelerometer-based wrist-worn device, Lin W-Y et al. quantified the regularity of PA of daily living, assessed the health condition, and monitored the sleep patterns of patients with COPD [22]. They also found that the PA of daily living, quality of life, and hospital readmission risk are strongly correlated, implying that PA data can be a strong predictor of 30-day hospital readmission [2,23,24]. Lin W.-Y. et al. [25] predicted 30-day hospital readmission using PA data and a conventional mathematical-statistical model with overall sensitivity of 62.96% and precision of 37.78%. They claimed that the low precision rate was due to unrecorded interventions, such as rehabilitation programs that prevented hospital readmissions, resulting in a high number of false predictions.

In a previous study using hospital medical claims data, knowledge-driven and data-driven features were evaluated, and an ML-based predictive model was proposed [26] to predict the readmission risk. They concluded that the performance (in terms of area under the receiver operating characteristic curve) improved from 0.60 to 0.653 after using a combination of knowledge-driven and data-driven features. Another study developed an ML-based 30-day readmission prediction model using a database of discharge abstracts, administrative claims data, patient characteristics such as age, sex, height, weight, and body mass index (BMI); and self-reported clinical indices, e.g., the smoking index, Barthel index for ADL, Hugh-Jones score, and Japan coma scale at hospitalization, to predict readmissions [27]. In a recent study, Zhou S.-M. et al. predicted hospital readmissions for campylobacteriosis using ML and text mining approach with 73% sensitivity and 54% specificity [28]. Hospitalization at triage was predicted using two deep learning methods, multi-layer perceptron (MLP) and convolutional neural network (CNN), conducted independently in parallel and classifier accuracy was area under the receiver operating characteristic curve ≈ 0.83 [29].

These existing methods predict readmissions using ML-based models from primarily self-reported, manually-assessed, and discontinuous data (e.g., demographics, initial vital signs and laboratory results, information extracted from self-reported ADL, past medical history and comorbidities, pre-stroke functional status as assessed using the modified Rankin scale); however, such methods are generally slow, inaccurate, and susceptible to human error. Most importantly, with such methods, readmissions were not predicted using the latest PA data. Thus, readmission prediction performance can be improved using latest continuous PA data and a ML-based predictive model.

Previous studies have extensively investigated readmission prediction using data from different sources, including PA data. However, to the best of our knowledge, no study has considered current PA data in daily life using predictive models. Thus, predicting 30-day hospital readmissions for patients with COPD in real-time is a challenging problem. In addition, ML-based predictive models with demographic and clinical data can also be used to predict pre-discharge of patients with COPD when readmitted to hospital [30,31].

Thus, in this study, we analyzed continuous PA data recorded using an accelerometer-based wrist-worn device and readmission information collected from hospital records to predict hospital readmissions. PA data from patients with COPD diagnosed and discharged from the hospital and living a normal life were used to train the ML model and predict the hospital readmissions in next 30 days. Our method either predicted or did not predict hospital readmission using the latest continuous PA data, and if a readmission is predicted, then it is either a true prediction or a false prediction. The remainder of this paper is organized as follows. A detailed description of the proposed method is given in the following section. We then present experimental results, a general discussion, and finally a conclusion.

## 2. Materials and Methods

### 2.1. Study Design and Setting

This study was conducted at Linkou Chang Gung Memorial Hospital, Taiwan (R.O.C.). We used PA data recorded with an accelerometer-based wearable device from Institutional Review Board (IRB) (approval number 104-6517B) and readmission information extracted from hospital records.

### 2.2. Participants

As shown in Figure 1, this study included 16 patients diagnosed with COPD and discharged from the hospital. Total data were collected over 3877 days, which included PA data and hospital readmission information, as reported previously [25].

### 2.3. Data Preprocessing, Statistical Analysis, and Data Reduction

Raw PA data were preprocessed using a statistical-mathematical model to perform data shrinkage and eliminate errors that occurred during data acquisition. Each raw data file was inspected for errors, e.g., missing values in the raw PA data, and to ensure that a minimum of 16 h of continuous PA data were recorded over each 24-h period. Table 1 lists the activity-based PA parameters obtained after applying statistical data processing techniques, resulting in activity-based parameters [22] that were used to monitor ADL and predict 30-day readmission using a conventional statistical model.

In this study, we organized the total data and data nomenclature was introduced for referring to a specific type of data, e.g., total data refers to a collection of entire available data, including the preprocessed PA data, activity-based PA parameters (AI, RI, QoA), statistical representative PA data values (mean, median, CoV), and actual readmission (RA) information from hospital records. However, total data recorded from 16 patients with COPD but the organized data resulted in 1695 datasets used for training and testing the ML models, interpret the prediction results, and evaluate the performance of the predictive models, therefore, available data is reasonably adequate to infer the results and draw conclusions. The data terms are defined in the following section.

### 2.4. Dataset

In this study, dataset refers to 7 days of continuous PA data and activity-based parameters derived from the PA data (Table 1).

#### 2.4.1. Valid Dataset

Total data, i.e., PA data and clinical information of patients with COPD, such as hospital readmissions, were collected and these two data collection processes ended at the same time. We could not validate the readmission predicted using the latest dataset; thus, we defined a valid dataset as any dataset for which actual hospital information was available for the next 30 days.

#### 2.4.2. Positive Dataset

For a valid dataset, if an actual hospital readmission event occurred in the subsequent 30-day period from the last day in the dataset, the concerned valid dataset was considered a positive dataset.

#### 2.4.3. Negative Dataset

For a valid dataset, if no actual hospital readmission event occurred in the subsequent 30-day period from the last day in the dataset, the concerned valid dataset was considered a negative dataset.

Physiological clinical information recorded at regular intervals was defined in data collection protocols, and a clinical follow-up was also performed during the same period. Here, we acquired patient characteristics and clinical information in the record, e.g., readmissions, emergency room visits, data retrieval, and device replacement.

The organized total data comprised 1695 datasets which were used in pipeline for the 10-fold cross-validation of the four ML models considered in this study, i.e., the logistic regression model, support vector machine, random forest model, and multilayer perceptron, to generate 30-day hospital readmission predictions by identifying the nonlinear classifying characteristic relationships among different activity-based PA parameters (Table 1) and actual hospital readmissions. We adopted 10-fold cross-validation (sometimes as blocked cross-validation for timeseries splits) to prevent any data leakage [32,33,34]. The ML models were trained by a combination of supervised and reinforced learning methods, and their performance was 10-fold cross-validated using the total dataset to acquire a final trained ML model with an averaged training score. This final trained model was used to test our method for generating 30-day hospital readmission predictions using the valid dataset. Among the four different ML models, the performance of the predictive model based on the logistic regression was found to be the best performing model with the PA data in terms of model performance metrics while other three models failed in the prediction criteria by having lower prediction accuracy (prediction-based), lower prediction precision (event-based) and higher false positive rates. A brief description of the regression method is given in the following section.

Usually, regression models are used to determine the strength and nature of the relationship between dependent variable Y and a number of independent variables X. The simplest form of regression is linear regression as shown in the Figure 2a; a mathematical representation of a linear regression model with a single X variable with its prediction error ε, and Figure 2b shows the bias and variance relationship of the regression model. A tradeoff approach was adopted to improve the model performance to minimize bias and variance and avoid overfitting and underfitting the model.

The performance of a logistic regression model is better with multiclass data labels, such as the datasets prepared using PA data in this study, compared with other types of models, e.g., support vector machine and random forest models. Thus, the logistic regression model is the most suitable for many neural network techniques.

As shown in Figure 3, the mathematical sigmoid function can be used to model and implement a generic logistic regression model, where x is the independent variable, and σ(x) is the respective predicted classification.

The algorithmic flow to model logistic regression is summarized as follows:

1: Start with random training weights: $w_{1},\;w_{2},\;\ldots ,\;w_{n},\;b$

2: for every point ($x_{1},\;x_{2},\;\ldots ,\;x_{n}$): do

3: for i=1, 2, …, n do

4: Update $w_{i}^{\prime }\;\leftarrow \;w_{i}-\alpha \left(\hat{y}-y\right)x_{i}$

5: Update $b^{\prime }\;\leftarrow b-\alpha \left(\hat{y}-y\right)$

6: Repeat until error is minimized

As shown in Figure 4, the data were divided into validation and training sets, where 10% of the data were used as the validation set, and the remaining 90% of the data were used as the training set. The ML model was trained and 10-fold cross-validated with an initial set of random weights, and these weights were updated to minimize the model error and improve performance metrics (or realize an effective tradeoff between error and performance) in each iteration. Moreover, in each iteration, the model was trained and validated using different groups of training and validation data, and the model’s loss function and parameters were tuned to minimize the loss function. The final validated model was taken as the average score used for testing, i.e., to generate readmission prediction with a valid dataset. Note that the same approach was repeated for the other ML models.

Figure 5 represents the retrospective approach used to prepare the testing datasets from the valid datasets. These testing datasets comprised PA data from the last 7 days. The testing datasets were given to the trained ML model to predict the hospital readmission in next 30 days, e.g., to predict the hospital readmission $P_{1}$ on 14 April 2016, Dataset-1 (continuous 7-days PA data from 7 April 2016 to 13 April 2016) is given to the trained-validated ML model.

In fact, on a concerned day, our method either predicted or did not predict hospital readmission; and if a hospital readmission is predicted, then it could be either a true prediction (TP) or a false prediction (FP) depending on whether the actual readmission event occurs in the next 30 days. For example, if any actual hospital readmission occurs before 13 May 2016, then $P_{1}$ is considered as a TP; else it is considered as an FP. Similarly, using Dataset-2 (continuous 7-days PA data from 8 April 2016 to 14 April 2016) hospital readmission on 15 April 2016, $P_{2}$, can be generated by the trained and validated ML model. Thus, hospital readmissions predicted using the overlapped testing datasets, i.e., readmission predicted according to the latest PA data, ensured real-time hospital readmission prediction in the next 30 days.

## 3. Results

This study included data from 16 patients with COPD collected over 3877 days. The PA data were collected along with clinical information of each patient with COPD from their hospital records. In total 1695 PA datasets were constructed, which were further reduced to 1361 valid datasets. Table 2 summarizes the total data collected in terms of demographic characteristics. The participants had a mean (SD) age of 74.0 (±11.2) years, with maximum and minimum ages of 94 and 54 years, respectively. In terms of the patients’ clinical characteristics, the mean (SD) height was 1.6 (±0.06) m, where the maximum and minimum heights were 1.71 and 1.5 m, respectively. The mean BMI was 21.96, with their median and inter-quartile range 5.70 to 24.98, respectively.

The final prediction performance was interpreted and summarized using two approaches, i.e., **prediction-based performance** and **event-based performance**, using the following performance metrics.

### 3.1. Prediction-Based Performance

#### 3.1.1. True Prediction (TP)

Using a valid dataset, a predicted readmission was considered a true prediction (TP) if any actual readmission occurred within the subsequent 30 days relative to the last day in the valid dataset.

#### 3.1.2. False Prediction (FP)

Using a valid dataset, a predicted readmission was considered a FP if no readmission occurred within the next 30 days relative to the last day in the valid dataset

### 3.2. Event-Based Performance

#### 3.2.1. Truly Predicted Event (TE)

An actual readmission event is considered as a truly predicted event (TE) if any readmission was predicted 30-day prior from the occurrence of actual readmission.

#### 3.2.2. Mispredicted Event (ME)

An actual readmission event is considered as a mispredicted event (ME) if no readmission was predicted within the 30-day period prior to the actual readmission.

Among the four ML models considered in this study, we found that the logistic regression-based ML model showed the best performance. Finally, the ML model’s performance was evaluated in terms of prediction-based and event-based performance, as expressed by Equations (1) and (2), respectively.
(1)$$Accuracy\;of\;prediction\;\left(Prediction\;based\right)=\frac{TP}{TP+FP}\times 100$$
(2)$$Precision\;\left(Event\;based\right)=\frac{TE}{TE+ME}\times 100$$

The total number of hospital readmission events was 21. In the event-based analysis, for the 21 hospital readmissions, 15 events were truly predicted, and 6 events were mispredicted. In other words, 71.43% of the readmission events were accurately predicted, and 28.57% of the readmission events could not be predicted. Note that some events were mispredicted due to an abrupt decline in the patient’s health condition, which could be reflected in the retrospective PA data and cannot be predicted.

In the prediction-based analysis, among the 1361 valid datasets, our method predicted hospital readmission for 199 valid datasets, and validation resulted in 140 predictions as true predictions and 59 predictions as false predictions. The prediction accuracy in this study is 70.35%, which represents a significant improvement, compared with the previously reported accuracy of 37.78% in [25] that was obtained using a conventional statistical model. Moreover, the false prediction rate was 29.65%, which has improved from the previously reported [25] false prediction rate of 62.22%.

To our knowledge, no research similar to this study, has been undertaken to predict 30-day hospital readmission using the latest continuous PA data that can be used for comparison purposes. Few studies have predicted 30-day hospital readmission using an index/score evaluated from prerecorded PA data, and even fewer studies have examined the effectiveness of PA for hospital readmission. 

Table 3 lists the performances of the different 30-day readmission prediction models used for the comparison. Compared with our previous reported [25] results of 21 actual hospital readmissions events, in the current study, the accuracy of the predicted events improved from 52.38% to 71.43% in the event-based analysis, and the prediction precision had improved from 37.78% to 70.35% in terms of the prediction-based analysis.

Amalakuhan et al. [5] employed a random forest-based ML model using demographic data and hospital records based on ICD-9 codes, and obtained a TP rate of 70%, which is nearly the same as that obtained in the current study. In addition, Chawla et al. [23] used prerecorded PA data and summarized the acceleration recorded in 3-planes (directions) to evaluate the vector magnitude units (VMU). Here, a logistic regression ML model was used for the prediction, which revealed that 31.58% of all patients experience readmission within 30 days of hospital discharge. They also found that patients with lower PA in daily life are 6.7 times more likely to be readmitted in the next 30 days. Hospital medical claims data were used by Min et al. [26] to derive two feature sets, i.e., knowledge-driven features and data-driven features. They found that using the knowledge-driven features, e.g., LACE index and hospital score, improved prediction performance from 65% to 65.3% using ML models. In another study, Goto et al. [27] used PA data with a logistic regression model with Lasso regression. They found that 7% of the patients experienced at least one 30-day hospital readmission with a precision classification ability of 61%, i.e., precision in prediction. Thus, our method has achieved significantly better prediction accuracy and precision compared with previous studies.

## 4. Discussion

Predicting hospital readmission for patients with COPD is a significant and challenging problem. Previous data collection and analysis methods were unable to predict readmissions precisely or prevent readmissions. In our analysis of the 30-day readmission prediction for patients with COPD, we encountered some hospital readmission cases that were easy to predict and some that were difficult. This may be limited to insufficient readmission events, resulting in a poorly trained ML model, or improper PA data values, which could occur for various reasons, e.g., the patient was not wearing the device, the device’s battery was low, or the patient did not follow the data collection protocols properly. To identify readmission records that are difficult to detect at the time of discharge for an initial admission, we must examine the PA data thoroughly using manual techniques or software tools to detect anomalies in the recorded data.

This study has certain limitations due to the data collection method which was not accurate as it had issues, such as nonreturn of some wearable device units out of scheduled protocols, occasional device malfunctions, low device battery power, and patients not wearing devices according to the requisite data recording protocols; thus, the desired PA measurements could not be obtained as expected from all participating patients with COPD.

## 5. Conclusions

This paper proposes a method to predict 30-day hospital readmissions for patients with COPD after discharge using PA data of daily life and ML models. In this study, activity-based PA parameters were evaluated by quantifying activities as acceleration-based PA data. Due to the simpler PA data collection method and corresponding ML model analysis, the overall 30-day readmission prediction performance gives encouraging results. To the best of our knowledge, no previous study has used latest continuous and dynamic PA data to predict readmissions in real-time. Most previous studies considered prerecorded PA data or static PA data to assess index as an auxiliary risk predictor.

Although the method proposed in this study uses latest continuous PA data to predict 30-day readmissions, if different patients with COPD exhibit similar PA data patterns and their predicted 30-day readmissions differ, this may result in prediction ambiguity because this method generates predictions by retrospectively considering the PA data retrospectively. Additionally, this prediction method can also be used to predict the hospital discharge of inpatients with COPD inpatients by monitoring their PA in the hospital. Thus, we believe that the proposed method can be applied in two unique applications, including the prediction of hospital readmission and discharge.

## Figures and Tables

**Figure 1 biosensors-12-00605-f001:**
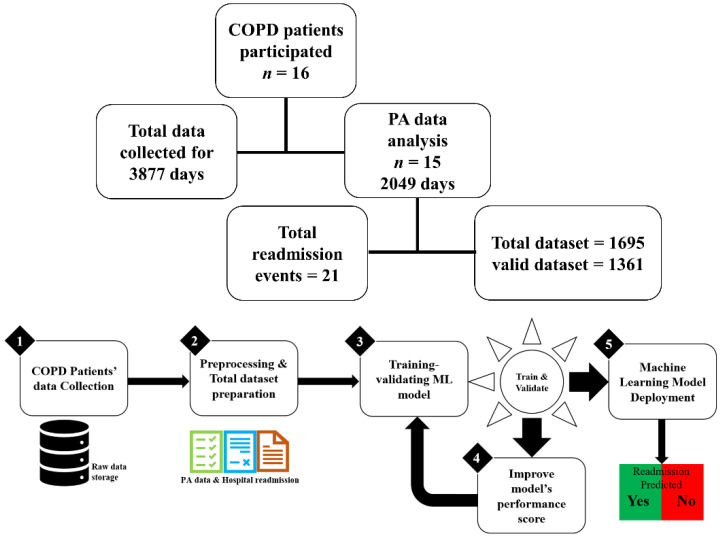
Patient data collection details and an overview of research methodology.

**Figure 2 biosensors-12-00605-f002:**
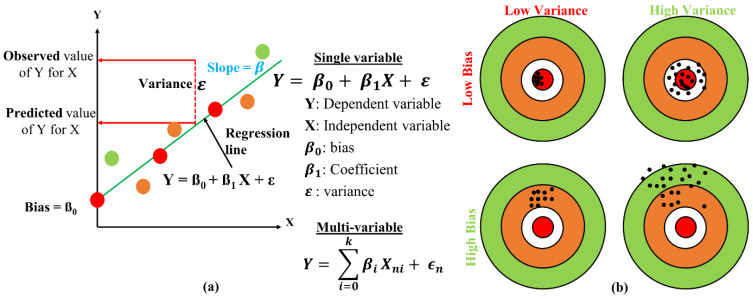
(**a**) Mathematical modeling of linear regression; (**b**) bias and variance.

**Figure 3 biosensors-12-00605-f003:**
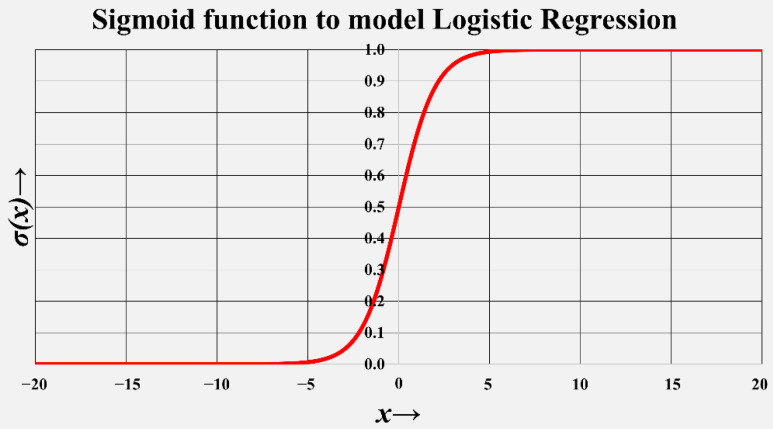
Sigmoid function to model logistic regression ML model.

**Figure 4 biosensors-12-00605-f004:**
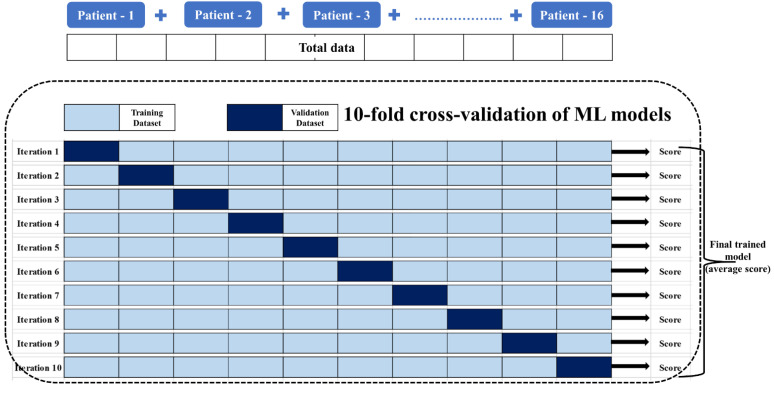
k — fold cross-validation of ML model, where k=10.

**Figure 5 biosensors-12-00605-f005:**
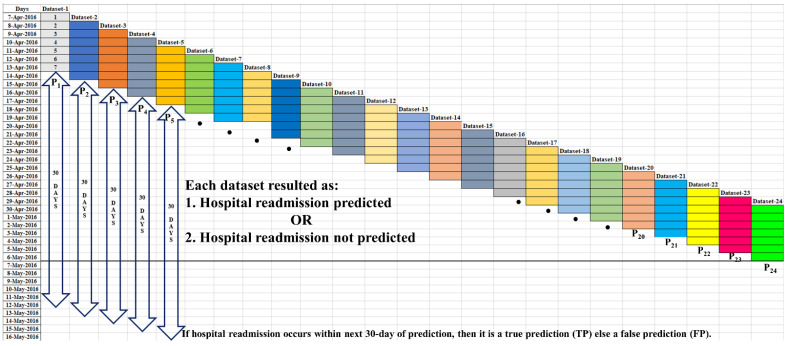
PA dataset method for 30-day hospital readmission prediction.

**Table 1 biosensors-12-00605-t001:** Conceptualized activity-based parameters in this study.

Name	Statistical Expression	Description
Resultant acceleration, $\;A_{i}$	$A_{i}=\sqrt{(a_{x\_i}^{2}+a_{y\_i}^{2}+a_{z\_i}^{2}}$	$a_{x\_i}$, $a_{y\_i}$, and $a_{z\_i}$ are accelerations along 3-orthogonal directions: the x−, y−, and z− axes at instance ”i”
Mean resultant acceleration, μ	$\mu =\frac{1}{N}\sum_{i=1}^{N}A_{i}$	The mean value μ is calculated over one epoch period (5 s) and N=100 values of $\;A_{i}$
Standard deviation (SD), $\;\sigma_{k}$	$\sigma_{k}=\sqrt{\frac{1}{N}\sum_{i=1}^{N}{\left(A_{i}-\mu \right)}^{2}}$	The SD value removes the constant gravity component included in the acceleration to represent actual quantified PA in an epoch period
Activity index, AI	$AI=\sum_{k=1}^{M}\sigma_{k}$	Summation of quantified PA for 12 epochs (total of 60 s for 5 s epoch)
Regularity index, RI	$RI=CORREL_{HH=12:00}^{HH=11:00}\left(\sum_{m=1}^{m=60}\left[AI_{d-1}\right],\sum_{m=1}^{m=60}\left[AI_{d}\right]\right)$	On day “*d*”, the regularity of the hourly-PA (RI) is defined as the correlation coefficient between the 24-h hourly-PA patterns of days “*d − 1*” and “*d*”.
Quality of physical activity, QoA	$QoA=\sum_{m=1}^{m=1440}AI_{m}\times \left(1+RI\right)$	Incorporating the effects of the different PAs and the regularity of PA on a day-to-day basis, ‘m’ can be 1~1440 min in a 24-h day

**Table 2 biosensors-12-00605-t002:** Data collection and results statistics.

Characteristics	Baseline Measure, *n* = 16
**Demographic**	
Age (years), mean (SD)	74.0 (±11.2)
**Clinical**	
Height (m), mean (SD)	1.60 (±0.06)
Body weight (Kg.), mean (SD)	55.39 (±9.01)
Body mass index (Kg/m^2^), median (IQR)	21.96 (5.70 to 24.98)
mMRC, mean (SD)	2.25 (±0.93)
6MWD, mean (SD)	282.56 (±98.10)
+FVC—Forced Vital Capacity Lung size (ltr), mean (SD)	1.72 (±0.47)
FVC—Forced Vital Capacity FVC(%), mean (SD)	59.56 (±16.05)
FEV—Forced expiratory volume FEV_1_(L), mean (SD)	0.81 (±0.27)
FEV—Forced expiratory volume FEV_1_(%), mean (SD)	38.25 (±15.68)
Tiffeneau-Pinelli index FEV_1_/FVC, mean (SD)	48.25 (±15.03)
**Study data statistics**	
Actual number of hospital readmissions	21
Total testing days	3877
Total datatsets	1695
Total valid datatsets	1361
Datasets with predictions	199
Datasets with true prediction (TP)	140
Datasets with false prediction (FP)	59
Truly predicted event (TE)	15
Mispredicted event (ME)	6

IQR: Interquartile range. SD: Standard deviation. mMRC: Modified Medical Research Council Dyspnea Scale. 6MWD: 6-min walk distance.

**Table 3 biosensors-12-00605-t003:** Performance of 30-day hospital readmission prediction models.

Studies	Methods	Data	Prediction Model Performance
Current study	PA data with a logistic regression ML model	Continuous PA data, hospital medical records	Accuracy of predicted events: 71.43%Precision (TP rate): 70.35%
Lin W.-Y. et al. [25]	PA data with statistical-mathematical model	Continuous PA data, hospital records	Accuracy of predicted events: 52.38%Precision (TP rate): 37.78%
Amalakuhan B. et al. [5]	55 feature variables for COPD exacerbations, random forest ML model	Demographic data, hospital medical records based on ICD-9 codes	Positive predictive value (accuracy in prediction): 70% (0.7)
Chawla H. et al. [23]	Vector magnitude units (VMU), i.e., summed movements in three planes over each minute, logistic regression ML model	PA data recorded with GT3X+ accelerometer, derived indices, hospital medical records	31.58% of patients had all-cause hospital readmissions, patients with lower PA are 6.7 times more likely to be readmitted
Min X. et al. [26]	Traditional and deep learning ML models: logistic regression, support vector machine, random forest, and multilayer perceptron	Knowledge-driven: hospital Score, LACE index, handcrafted features; Data-driven: reshaped data grouped into categories	Prediction performance with data-driven features: 65% Combined (knowledge-driven and data-driven): 65.30%
Goto T. et al. [27]	Recorded PA data used with logistic regression and Lasso regression ML models	Self-reported, manually assessed, static PA data	7% of patients had 30-day readmissions. Prediction classification ability (precision): 61.00%

## Data Availability

Data are available from the corresponding author upon reasonable request.

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
