# Peer review of "Machine Learning-Based 30-Day Hospital Readmission Predictions for COPD Patients Using Physical Activity Data of Daily Living with Accelerometer-Based Device"

_biosensors, 2022, doi:10.3390/bios12080605_

Round 1
Reviewer 1 Report
This manuscript presented a study that aimed to apply machine learning and continuous physical activity (PA) data to predict hospital readmission within the next 30 days for COPD patients. Please see below my comments for the authors to consider and address.
1. In “I. Introduction”, the 2nd sentence: would the authors add more clarifications about the “other disease readmissions” under the context of reference 2?
2. In “Materials and Methods”, would the authors add more details about the data splitting and labeling? For example,
a. How were the training sets and the testing sets determined?
b. What was the difference between the testing datasets illustrated in Figure 5 and the testing datasets used for model validation? (See page 7, the 1st paragraph, “The testing datasets were input to the trained and validated ML model…)
c. And more as the authors see fit to provide a clearer method description on data preparation.
3. On page 5, would the authors add contents describing more about the performance of the other three models? Including performance stats used for model selection would be better.
4. The number of patients enrolled in this study was 16, which seems to be a smaller number compared to some cited studies. Would the authors add more discussions on how the size of datasets and the number of enrolled patients in this study compared to others, and how the number of patients enrolled would impact the evaluation and performance of the built model?
5. How was this study improved or different from the one previously described in 10.1109/SMC42975.2020.9283295?
6. How did the trained and validated model perform in random datasets or datasets that were from non-COPD patients?
Author Response
Manuscript ID: biosensors-1788672
Manuscript title: Machine Learning-based 30-day hospital readmission predictions for COPD patients using Physical Activity data of daily living with Accelerometer-based device
Response to Reviewer 1 Comments
We sincerely thank Reviewer 1 for the time and consideration given in the reviewing our manuscript. We also thank for the reviewer’s comments and we have carefully gone through the reviewer’s comments. The detailed responses to the reviewer’s comment are listed below and parts of the responses are also added in the revised manuscript as per the suggestion.
Comment #1: In “I. Introduction”, the 2nd sentence: would the authors add more clarifications about the “other disease readmissions” under the context of reference 2?
Authors’ response: We thank the reviewer for this comment and drawing our attention for clarifying this sentence. We recognize that this sentence was not clear and revised the manuscript as: “Patients with COPD have the highest number of 30-day hospital readmissions compared with other diseases, such as pneumonia, gastrointestinal disorders, heart failure, and other respiratory/ ventilatory diseases [2]”. Here, “other disease readmissions” mean 30-day hospital readmissions due to other chronic diseases such as pneumonia, gastrointestinal disorders, heart failure, other respiratory/ ventilatory diseases, etc. as described in the Fig. 3 of the cited reference [2].
Comment #2: In “Materials and Methods”, would the authors add more details about the data splitting and labeling? For example,
- How were the training sets and the testing sets determined?
- What was the difference between the testing datasets illustrated in Figure 5 and the testing datasets used for model validation? (See page 7, the 1st paragraph, “The testing datasets were input to the trained and validated ML model…)
- And more as the authors see fit to provide a clearer method description on data preparation.
Authors’ response: We thank the reviewer for this insightful comment and raising this very interesting point. We revised the “Materials and Methods” section as per reviewer’s suggestion to give clear understanding of data splitting and labeling. Point-wise responses to the comments are as follows:
- After organizing the total data, training sets were formed using PA data and activity-based PA parameters derived from the PA data (as listed in Table 1, revised manuscript) along with hospital readmission data extracted from patient’s record. Testing sets were formed using ‘dataset’ refers to 7 days of continuous PA data and it is clearly explained in the revised manuscript illustrated in Fig. 5.
- Testing dataset illustrated in Fig. 5 refers to 7 days of continuous PA data. For model validation, we used 10-fold cross-validation method for 4 ML models—logistic regression, random forest, support vector machine, multilayer perceptron—with training dataset and validation dataset; the best performing model, i.e. logistic regression in our case, is selected and tested as described above in response
- We apologize for this inconvenience in understanding, “Materials and Methods” section in the revised manuscript we have explained terms used and data preparation more clearly as per reviewer’s suggestion.
Comment #3: On page 5, would the authors add contents describing more about the performance of the other three models? Including performance stats used for model selection would be better.
Authors’ response: We thank the reviewer and clearly understand this important concern. As per reviewer’s suggestion, we briefly discussed about the performance of other 3 ML models in revised manuscript as: “Among the four different ML models, the performance of the predictive model based on the logistic regression was found to be the best performing model with the PA data in terms of model performance metrics while other three models failed in the prediction criteria by having lower prediction accuracy (prediction-based), lower prediction precision (event-based) and higher false positive rates.” Other studies summarized final results as a percentage of total readmissions or their model’s prediction performance ability (listed in the Table 3, revised manuscript), but our study was aimed to actually predict a readmission event using 4 ML models in which 3 models failed in the prediction criteria by having lower prediction accuracy (prediction-based), lower prediction precision (event-based) and higher false positive rates.
However, the aim and scope of this manuscript is to present our prediction methodology and report the best performing model, so we have considered reviewer’s this suggestion positively and, in the future, we will be presenting a well-structured performance comparison of different ML models.
Comment #4: The number of patients enrolled in this study was 16, which seems to be a smaller number compared to some cited studies. Would the authors add more discussions on how the size of datasets and the number of enrolled patients in this study compared to others, and how the number of patients enrolled would impact the evaluation and performance of the built model?
Authors’ response: We are delighted and thankful to the reviewer for raising this point and giving a chance to revise our manuscript accordingly as: “However, total data were from 16 patients with COPD but the organized data resulted in 1,695 datasets used for training and testing the ML models, interpret the prediction results, and evaluate the performance of the predictive models which is reasonably adequate to infer the results and draw conclusions.
As a matter of fact, we have collected data for 3,877 days in which data for 2,049 days were considered for data analysis. As explained above in the response to Comment #2 a., we have envisaged and defined dataset which were actually considered, so following this approach we had 1,695 datasets which is reasonably adequate to infer and drawing conclusions.
Comment #5: How was this study improved or different from the one previously described in 10.1109/SMC42975.2020.9283295?
Authors’ response: We thank reviewer for mentioning our international conference publication in 2020 IEEE International Conference on Systems, Man, and Cybernetics (SMC) which reported our preliminary results of 30-day hospital readmission prediction using ML model with PA data. In this paper, we reported our prediction results with sensitivity 0.88 and positive predictive value 0.75 but our results were limited due to higher false positive rates. In this revised manuscript, we are reporting a rectified analysis approach with significantly improved results, like lower false positive rates as better as 29.65%.
Comment #6: How did the trained and validated model perform in random datasets or datasets that were from non-COPD patients?
Authors’ response: We are thankful to the reviewer for giving this valuable suggestion, we will surely analyze this in future. The work presented in this manuscript is limited to the data only from patients with COPD.
Authors' final statement: We sincerely thank Reviewer 1 for the time and consideration given in the reviewing our manuscript.

Reviewer 2 Report
I would like to thank the authors very much for this work. The methodology is mostly clear, and the manuscript is well-written as well. It is also good to see that the authors are aware of possible limitations. In general, I believe that the study has a good potential. However, there are some points that should be considered, please.
(1)
Most importantly, it is not entirely clear how the dataset was split into train and test sets. I am wondering if the train set included PA data from multiple participants? If so, I am afraid that this would be considered as a case of ‘information leakage’, and the model performance should be re-visited.
(2)
Regarding the related work, there is a dire need to refer to more recent contributions that applied the state-of the-art ML techniques to predict hospital admission/ readmission. For example:
https://doi.org/10.1109/BigData50022.2020.9378073
https://doi.org/10.3390/jpm12010086
(3)
In my view, the limitations should also discuss the dataset size, which was based on 16 participants only.
Author Response
Manuscript ID: biosensors-1788672
Manuscript title: Machine Learning-based 30-day hospital readmission predictions for COPD patients using Physical Activity data of daily living with Accelerometer-based device
Response to the Reviewer 2 Comments
We are highly thankful to the reviewer 2 for the comments and positive affirmation given to our work reported in this manuscript. We thank reviewer 2 for the time, effort, and consideration in reviewing our manuscript. Responses to the reviewer’s comment can be found below and parts of the responses are also added in the revised manuscript.
Comment #1: Most importantly, it is not entirely clear how the dataset was split into train and test sets. I am wondering if the train set included PA data from multiple participants? If so, I am afraid that this would be considered as a case of ‘information leakage’, and the model performance should be re-visited.
Authors’ response: We thank reviewer for giving this important comment. We have revised the manuscript as per reviewer’s suggestion in “Materials and Methods” section to clearly explain about the data preparation. We have adopted following points and avoided data (information) leakage, the same have also been added in the revised manuscript:
- We used pipeline data for 10-fold cross-validation which prevents data (information) leakage and provides a better estimate of the model's performance on unseen data as discussed in the cited references in revised manuscript [33, 34].
- Also, in our earlier reported statistical-mathematical model-based prediction, we have conceptualized and established that — “Health status is a function of PA data, and declining PA of daily living indicates poor health status and therefore, hospital readmission”— in this we had analyzed each COPD patients' data individually, but we applied the same readmission prediction criteria on each patient's PA data. This concluded that the hospital readmission prediction depends upon the properties of PA data only not upon the specific patient.
- In short, training data included shuffled PA data and testing data included latest 7-days of PA data (dataset), which ensured that the testing data cannot be as same as training data, hence preventing any data (information) leakage.
Comment #2: Regarding the related work, there is a dire need to refer to more recent contributions that applied the state-of the-art ML techniques to predict hospital admission/ readmission. For example:
https://doi.org/10.1109/BigData50022.2020.9378073
https://doi.org/10.3390/jpm12010086.
Authors’ response: We are thankful to the reviewer for suggesting these recent contributions which further helped us to present our idea. Also, same have been cited in the “Introduction” section of the revised manuscript to better explain our idea and strengthen our approach.
Manuscript has been revised as per reviewer’s suggestion as: “In a recent study, Zhou S-M et al. predicted hospital readmissions for campylobacteriosis using ML and text mining approach with 73% sensitivity and 54% specificity [28]. Hospitalization at triage was predicted using two deep learning methods, multi-layer perceptron (MLP) and convolutional neural network (CNN), conducted independently in parallel and classifier accuracy was area under the receiver operating characteristic curve ≈ 0.83 [29].”
Comment #3: In my view, the limitations should also discuss the dataset size, which was based on 16 participants only.
Authors’ response: We respect reviewer’s view and delighted that the reviewer has raised a valid point. As a matter of fact, we have collected data for 3,877 days in which data for 2,049 days were considered for data analysis. As explained above in the response to Reviewer 1 Comment #2 a., we have envisaged and defined dataset which were actually considered, so following this approach we had 1,695 datasets which is reasonably adequate to infer and drawing conclusions.
Authors' final statement: We sincerely thank Reviewer 2 for the time and consideration given in the reviewing our manuscript.

Round 2
Reviewer 1 Report
Thank you for addressing the previous comments. I do not have additional comments besides please checking for typos such as the "[3234]" on line 181, page 5 of 14, which probably is "[32-34]".
Author Response
Manuscript ID: biosensors-1788672
Manuscript title: Machine Learning-based 30-day hospital readmission predictions for COPD patients using Physical Activity data of daily living with Accelerometer-based device
Response to Reviewer 1 Comments
We sincerely thank Reviewer 1 for giving the time and consideration once again in reviewing our revised manuscript. We have carefully gone through the manuscript for any typos and minor errors.
Comment #1: Thank you for addressing the previous comments. I do not have additional comments besides please checking for typos such as the "[3234]" on line 181, page 5 of 14, which probably is "[32-34]".
Authors’ response: We thank the reviewer for drawing our attention towards this typo error which has been corrected now.
Authors' final statement: We sincerely thank Reviewer 1 for the time and consideration given in the reviewing our revised manuscript.

Reviewer 2 Report
Thanks very much for accommodating the feedback. I have no further comments.
Author Response
Manuscript ID: biosensors-1788672
Manuscript title: Machine Learning-based 30-day hospital readmission predictions for COPD patients using Physical Activity data of daily living with Accelerometer-based device
Response to the Reviewer 2 Comments
We are highly thankful to the reviewer 2 for giving the time once again to review our revised manuscript.
Comment #1: Thanks very much for accommodating the feedback. I have no further comments.
Authors’ response: We thank reviewer for giving time and consideration, also, we have further corrected few typos and English for better understanding.
Authors' final statement: We sincerely thank Reviewer 2 for reviewing our revised manuscript.
